# Short-Term Effects of Dietary Protein Supplementation on Physical Recovery in Older Patients at Risk of Malnutrition during Inpatient Rehabilitation: A Pilot, Randomized, Controlled Trial

**DOI:** 10.3390/healthcare11162317

**Published:** 2023-08-17

**Authors:** Barbara Strasser, Vincent Grote, Walter Bily, Helena Nics, Patricia Riedl, Ines Jira, Michael J. Fischer

**Affiliations:** 1Ludwig Boltzmann Institute for Rehabilitation Research, A-1100 Vienna, Austria; 2Medical Faculty, Sigmund Freud Private University, A-1020 Vienna, Austria; 3Department of Physical Medicine and Rehabilitation, Ottakring Clinic, Vienna Health Association, A-1160 Vienna, Austria; 4Department of Physical Medicine and Rehabilitation, Floridsdorf Clinic, Vienna Health Association, A-1210 Vienna, Austria; 5VAMED Rehabilitation Center Kitzbuehel, A-6370 Kitzbuehel, Austria

**Keywords:** aging, dietary protein, hand-grip strength, physical function, rehabilitation, sarcopenia

## Abstract

It is estimated that about 50% of geriatric rehabilitation patients suffer from sarcopenia. Thereby, malnutrition is frequently associated with sarcopenia, and dietary intake is the main modifiable risk factor. During hospitalization, older adults are recommended to consume more dietary protein than the current recommended dietary allowance of 0.8 g/kg body weight per day to optimize the recovery of muscular strength and physical function. This prospective pilot study examined the feasibility and preliminary efficacy of short-term protein supplementation with protein-enriched foods and drinks on the hand-grip strength, nutritional status, and physical function of older patients at risk of malnutrition during a three-week inpatient orthopedic rehabilitation stay. The Mini Nutritional Assessment (MNA) tool was used to assess malnutrition. Patients with an MNA score ≤ 23.5 points were randomly assigned to an intervention group (goal: to consume 1.2–1.5 g protein/kg body weight per day) or a control group (standard care). Both groups carried out the same rehabilitation program. Physical recovery parameters were determined at admission and discharge. A trend was recognized for participants in the intervention group to consume more protein than the control group (*p* = 0.058): 95.3 (SD 13.2) g/day as compared to 77.2 (SD 24.2) g/day, which corresponds to a mean protein intake of 1.6 (SD 0.3) g/kg/day vs. 1.3 (SD 0.5) g/kg/day. Dietary protein supplementation increased body weight by an average of 0.9 (SD 1.1) kg and fat mass by an average of 0.9 (SD 1.2) kg as compared to the baseline (*p* = 0.039 and *p* = 0.050, respectively). No significant change in hand-grip strength, body composition, or physical function was observed. In conclusion, short-term intervention with protein-enriched foods and drinks enabled older patients at risk of malnutrition to increase their protein intake to levels that are higher than their required intake. In these older individuals with appropriate protein intake, dietary protein supplementation did not result in a greater improvement in physical recovery outcomes during short-term inpatient rehabilitation. The intervention improved dietary protein intake, but further research (e.g., a full-scale, randomized, controlled trial with sufficient power) is required to determine the effects on physical function outcomes.

## 1. Introduction

One of the many threats to an independent lifestyle is the age-related loss of muscle mass and strength, which is referred to as sarcopenia. Sarcopenia can lead to functional impairments and mobility limitations that are related to other geriatric syndromes, such as a propensity to experience falls and immobility [1]. Another important health risk in old age that is often poorly recognized and underdiagnosed is malnutrition. Malnutrition is frequently associated with sarcopenia and poorer chances of functional recovery [2,3]. Both conditions are highly prevalent in geriatric rehabilitation inpatients [4]; thus, they are important to address in the quest to prevent physical dependence in old age. Therefore, it is essential to diagnose and treat malnutrition during inpatient rehabilitation.

Initiating early treatments to maintain proper muscle mass and function is crucial to ensuring optimal patient outcomes across the healthcare continuum [5]. Interventions to support physical function and recovery in geriatric rehabilitation patients include resistance training and nutrition because both have been shown to improve muscular strength, body composition, and functional performance in older adults [6,7,8]. The currently recommended dietary allowance (RDA) for protein, 0.8 g protein per kilogram of body weight per day, might not be adequate for maintaining muscle health in old age. For this reason, experts have proposed increasing dietary protein recommendations for older age groups to 1.0 to 1.2 g/kg body weight per day, and an even higher protein intake (1.2 to 1.5 g/kg body weight/day) is advised for those who are exercising or for older people during hospitalization or rehabilitation [9].

The majority of hospital inpatients fail to meet even their minimum estimated energy and protein requirements [10,11,12]. The aim of this pilot study was to evaluate the feasibility (including recruitment, intervention uptake, completion of the post-intervention assessment, and completeness of outcome data collection) and the preliminary efficacy of short-term protein supplementation on physical recovery in older patients at risk of malnutrition during a three-week inpatient orthopedic rehabilitation stay. 

## 2. Material and Methods

### 2.1. Study Design and Participants

This study was designed as a single-center, randomized, controlled, open-label, parallel-group intervention pilot study with pre- and post-intervention assessments. The goal was to include 20 malnourished older patients who had been admitted to a three-week inpatient orthopedic rehabilitation at the Rehabilitation Center Kitzbuehel in Austria. All patients who were 65 years and older and were at risk of malnutrition were screened for study eligibility. Subjects were excluded if they had a food allergy or intolerance that restricted them from receiving the protein-rich menu or the protein-enriched intervention products, suffered from chronic renal insufficiency (stages 3 or 4), cognitive impairment, or had any other relevant medical history that prevented their participation in the intervention or could affect the study outcome. Eligible patients were asked to participate in this study within the first two days of their rehabilitation stay and signed a written informed consent after receiving detailed explanation of this study and its potential risks. The Medical Ethics Research Committee of Innsbruck Medical University gave their approval for this study.

### 2.2. Diagnosis of Malnutrition

The nutrition status of all participants was assessed at admission using the Mini Nutrition Assessment Short Form (MNA-SF) questionnaire [13]. If the participant’s score was 11 or less, indicating a “malnutrition risk”, a trained dietitian continued to ask the remaining questions to obtain additional information about factors that could impact their nutritional status. A score between 17 to 23.5 points indicated that the participant was “at risk of malnutrition”, and a score of fewer than 17 points indicated that they were “malnourished”. 

### 2.3. Nutritional Intervention 

Eligible participants were randomly assigned to an intervention group (*n* = 10) or a control group (*n* = 10) in a 1:1 ratio by using the Randomizer for Clinical Trials tool developed at the Medical University of Graz (http://www.randomizer.at/, accessed on 11 July 2023). During the rehabilitation stay, patients in both groups were free to choose different menus. Members of the intervention group received additional protein-enriched foods (goal: to consume 1.2 to 1.5 g/kg body weight/day) and a protein-enriched drink (providing 150 kcal and 10 g protein per serving, taken twice daily as a between-meal snack). To reach this goal, an individual nutritional plan was developed by a trained dietitian for each patient. The control group received the standard energy and protein-rich hospital menu that was developed for patients aged 65 years and older admitted to this rehabilitation center. 

### 2.4. Rehabilitation Program 

In Austria, an insured person who needs post-acute care or conservative treatments is eligible to receive medical rehabilitation care. The orthopedic rehabilitation program (WHO phase 2) lasts 21 days and comprises a variety of services, including exercise therapy (muscle-strengthening exercises for the hip, thigh, and upper arm and shoulder muscles on 3 non-consecutive days of the week, 30 min per session), electrotherapy, lymphatic drainage, massage, and hydrotherapy. Patients receive an average of 2 to 3 hours of treatment daily or 1800 therapy minutes in total during the 3-week program [14].

### 2.5. Outcomes and Data Collection

Data on participant recruitment rate, intervention uptake, and completeness of outcome data collection were recorded for feasibility outcome reporting. Nutritional intake was measured on day four and subsequently, in weeks two and three of the rehabilitation stay using a detailed three-day dietary protocol. A checklist of specific food and beverages was used to verify the reported intake, and a visual guide to portion sizes was used to estimate the portion sizes. The verified food records were entered into the food calculation program Necta (Evoca Group, Pinkafeld, Austria) to assess energy and protein intake according to the German Nutrient Database (BLS, Federal Research Centre for Nutrition and Food, Karlsruhe, Germany). In addition, analytical values compiled from food-producing firms were used to estimate the protein content of foods. Average energy and protein intakes were calculated for each patient during the rehabilitation stay. To estimate the patients’ nutritional needs, energy requirements were calculated based on the resting energy expenditure using the Harris and Benedict equation [15] and were multiplied by a factor of 1.4 to estimate the minimal energy requirements [16]. 

Maximum hand-grip strength (kg) was determined at admission and one day before discharge using a SAEHAN hand dynamometer (Saehan Corporation, Masan-si, Republic of Korea); the highest of the three measurements was reported for the dominant hand. The assessment of hand-grip strength has been assigned considerable clinical value, and this strength is considered a key characteristic of sarcopenia, with low hand-grip strength (<27 kg for men and < 16 kg for women) representing the first defining characteristic [17]. 

Measures of nutritional status included the MNA score at admission, prealbumin level (mg/dL), and the anthropometric and body composition parameters. The body weight, rounded up to the nearest 0.1 kg, was measured on a calibrated weighing chair without shoes or heavy clothing (KERN MCC 250K100M, Stuttgart-Balingen, Germany). The standing height, rounded up to the nearest 0.1 cm, was measured without footwear. The body mass index (BMI) was calculated by dividing the body weight by the height squared (kg/m^2^). Prior to and at the end of the intervention, all patients were tested for fat-free mass (kg), body cell mass (kg), and fat mass (kg) by using the bioelectrical impedance analysis (BIA) method (BIACORPUS RX4004M, MEDI CAL HealthCare GmbH, Karlsruhe, Germany). Furthermore, the BIA phase angle (PhA°) was determined as an index of the ratio between extracellular and intracellular water, body cell mass, and cellular integrity. A low PhA (cut-off point values from 4.05 to 5.05°) has been shown to be associated with sarcopenia and malnutrition and to be a predictive factor for hospitalization, falls, and frailty [18,19]. A body composition assessment was carried out by trained dietitians according to standard operating procedures. 

Each patient’s functional status was assessed with the Health Assessment Questionnaire (HAQ) [20]. The HAQ comprises 20 questions and has been widely validated. The score ranges from 0 (no functional limitations) to 3 (serious function limitations); a score below 0.5 is considered normal, whereas a score above 1.5 indicates severe disability. 

### 2.6. Sample Size 

In this pilot (phase 0), randomized, controlled trial, a sample size of twenty patients was considered to evaluate feasibility and limited efficacy testing in preparation for a larger/full-scale trial [21]. 

### 2.7. Statistical Analysis 

Descriptive statistics were used to describe baseline characteristics and are presented as means and standard deviation (SD). A per-protocol analysis was performed with SPSS (IBM SPSS Statistics version 27, IBM Corp, Armonk, NY, USA). Differences between the two groups at admission (t1) were examined with independent-samples *t*-test and χ^2^ test. Changes during rehabilitation from t1 to discharge (t2) were examined using dependent *t*-test and repeated-measures analyses of variance (rANOVA). To examine changes in variables between the intervention and control group during rehabilitation, the grouping variable was added to the rANOVA. Due to the small sample size and the exploratory nature of this pilot study, inferential statistics were performed to estimate the effects of the intervention (i.e., limited efficacy testing), and the statistical calculations performed are ultimately descriptive. When interpreting the observed effects, the empirical results from the literature were taken into consideration. *p*-values, therefore, express the replicability of the obtained results in the population under study. A *p*-value of less than 0.10 (two-sided test) was considered to indicate a trend toward statistical significance to reduce the beta risk.

## 3. Results

The baseline characteristics are presented in Table 1. Figure 1 shows the flow chart of the subjects with reasons for exclusion. Out of the 327 subjects assessed for eligibility, 163 subjects were not at risk of malnutrition. Regarding the 83 subjects assessed as being at risk of malnutrition, the physician did not refer patients to a dietitian for additional nutritional assessments and interventions. For another 54 subjects, the full MNA score indicated a good nutritional status. Of the 27 patients approached for recruitment, 20 agreed and 7 declined, yielding a recruitment rate of 74%. In total, twenty participants were randomized to either an intervention group (*n* = 10) or a control group (*n* = 10). The participant retention rate was 100% in terms of participants who were randomized to an intervention, attended study visits, and completed measures. At admission, all patients were at risk of malnutrition (MNA score: 21.1 (SD 1.9) points) with no significant differences noted between groups (*p* = 0.575). During rehabilitation, the intervention group consumed more protein than the control group (*p* = 0.058): 95.3 (SD 13.2) g/day as compared to 77.2 (SD 24.2) g/day, which corresponds to a mean protein intake of 1.6 (SD 0.3) g/kg per day vs. 1.3 (SD 0.5) g/kg per day. Moreover, the intervention group reached a higher energy intake than the control group (*p* = 0.021): 2180 (SD 385) kcal/day as compared to 1746 (SD 381) kcal/day. 

Figure 2 shows the average protein distribution across the self-selected main meals as a percent of the total provided amount of protein. Breakfast, lunch, and dinner provided 37%, 38%, and 25% protein by a whole food normal diet (Figure 2A) and 38%, 30%, and 32% protein by a whole food plant-based diet, respectively (Figure 2B).

Physical recovery outcomes are shown in Table 2. Body weight improved by an average of 0.9 (SD 1.1) kg in the intervention group (*p* = 0.039) but was not different between groups (*p* = 0.762; time × group: *p* = 0.487). The fat mass increased in participants given higher amounts of protein (*p* = 0.050) but did not change in the control group (*p* = 0.923; time × group: *p* = 0.110). A trend towards an increase in serum prealbumin was observed in the intervention group, ranging from a mean of 21.1 (SD 8.1) mg/dL at admission to 23.6 (SD 8.9) mg/dL at discharge (*p* = 0.070), whereas the serum prealbumin levels remained unchanged in the control group (*p* = 0.634; time × group: *p* = 0.092). In both groups, the hand-grip strength, fat-free mass, body cell mass, and physical function (HAQ) score did not change over time. 

## 4. Discussion

The European Society for Clinical Nutrition and Metabolism (ESPEN) guidelines on clinical nutrition and hydration in geriatrics recommend that a positive malnutrition screening should be followed by a systematic assessment, individual nutritional intervention, monitoring, and a corresponding adjustment of interventions [22]. Because most physicians are not trained to complete a comprehensive nutritional assessment, a critical function of the physician is working with dietitians and other health professionals (e.g., nurses and physical therapists) to implement nutrition care processes. In the present pilot study, almost 50% of the at-risk patients were not referred to a dietitian for further treatment. On the other hand, the retention rate (proportion of participants with valid dietary intake and physical recovery outcome data at follow-up) was found to be 100%. This highlights the importance of providing adequate clinical nutrition training for all healthcare professionals, including physicians, as a first step, as this enables them to provide timely and adequate nutritional support during rehabilitation stays for patients at risk of malnutrition. The next step would be to draw up a nutritional care plan using a multidisciplinary approach to ensure that older adults are assessed and treated accordingly to improve patient nutritional condition. 

Protein-enriched foods and drinks were successfully implemented in the menu of the older adults during inpatient rehabilitation. Although dietary protein supplementation increased the protein intake to levels that are higher than their required intake, the intervention did not result in a greater improvement in physical recovery outcomes. Time effects were identified for nutritional status (body weight and fat mass), and a trend toward statistical significance for prealbumin was observed. Overall, the average values for some outcomes increased in the intervention group, but few of these reached statistical significance and can be described as trends at best. This may be for the following reasons: (1) this study was not adequately powered, and a larger sample size is likely required to detect significant differences; (2) the high protein intake in the control group of 1.3 g/kg per day; and (3) the short follow-up time. The latter two explanations could also be associated with the fact that merely increasing protein might not be sufficient unless skeletal muscle mass is also built up, an especially relevant aspect for patients at risk of malnutrition [23,24]. Similarly, the results of a recent meta-analysis of randomized clinical trials demonstrated that although nutritional therapy increased daily energy and protein intake, nutritional support had few effects on functional outcomes in malnourished medical inpatients. These results, however, may be attributed to the relatively short duration of the nutritional support [25]. Because we did not want to act against the standard advice to consume a protein-rich diet, the rather high protein intake seen in the control group is probably due to the fact that the rehabilitation center offers protein- and energy-rich menus to older patients during rehabilitation. However, even the recommended high protein intake of 1.2 to 1.5 g/kg per day may be too low for functionally limited older patients to successfully recover muscular strength and physical function, especially those with chronic diseases [26]. 

Furthermore, the nature of the training stimulus using physiotherapy exercises may have been too low in intensity to stimulate a robust increase in muscle protein synthesis rates. Indeed, the training component per se is of primary importance in improving muscle mass and strength, as well as functional capacity, because a substantial proportion of the older population benefits from a resistance-type exercise training intervention [27]. However, the level of responsiveness to resistance training is strongly affected by the duration of the exercise intervention, with more positive responses observed after more prolonged exercise training [28].

In the present pilot study, older patients at risk of malnutrition exhibited a higher probability of sarcopenia, reaching a prevalence of 45%. This was indicated by their maximum hand-grip strength, which was compared with the cut-off point values for weak hand-grip strength obtained from a healthy, non-frail older population [29]. In a recent systematic review of data from a total of 34,955 participants older than 60 years, the prevalence of sarcopenia was approximately 10% in community-dwelling individuals, 24% in hospitalized individuals, and 51% and 31% for men and women, respectively, in nursing homes [30]. Thus, a significant proportion of older persons suffer from sarcopenia (i.e., a major proportion in clinical and nursing home settings) even in healthy populations. Sarcopenia is associated with adverse health outcomes, such as falls, fractures, functional impairments, and mobility limitations, and accompanied by an elevated risk for hospitalization, morbidity, and mortality [31]. 

Conversely, being at nutritional risk is significantly associated with odds of suffering from sarcopenia that are two- to three-times higher than normal, resulting in the invention of a new term: “sarcopenia malnutrition syndrome” [32]. The simultaneous presence of malnutrition has been shown to reinforce the loss of muscle mass, muscle strength, and function, which has severe implications for physical performance in older people in both the community and hospital settings [33,34,35]. An analysis of prevalence data for malnutrition and nutritional risk in older adults across different healthcare settings using MNA showed a wide range of malnutrition, ranging from 3% in the community setting to approximately 30% in rehabilitation and sub-acute care settings [36]. Additionally, nutritional status can be assessed by measuring BIA-derived PhA as a proxy for water distribution and body cell mass and, from a practical standpoint, as an index of overall muscle quality [17]. PhA decreases with malnutrition and is directly related to sarcopenia [18]. However, a low PhA mainly indicates an increased risk of malnutrition and does not reveal the underlying cause. It is safe to say, however, that PhA increases when resistance training is practiced and decreases when detraining or inflammation occurs [19]. We studied PhA in older rehabilitation patients and found low PhA levels (mean values from 3.83 ± 0.80°), observing no change over time; this may be due to the relatively short follow-up times accompanied by the unchanged lean body mass. A low PhA has been linked to dysmobility syndrome (osteoporosis, low lean mass, falls in the preceding year, low grip strength, high-fat mass, and poor timed up-and-go performance) [37], an increased risk of falls, and incident disability in older adults [38,39].

The purpose of this pilot study was to assess the feasibility and preliminary effectiveness of an individualized nutrition intervention in older patients at risk of malnutrition during inpatient rehabilitation. A major strength of this trial is the randomized, controlled study design, the inclusion of old and very old subjects, and the use of an objective and standardized test to assess muscle strength and function. However, several limitations must be considered when interpreting the data. Due to the low sample size (lack of power), as only 20 total participants were included in the analysis, this pilot study only provides an indication of its possible effects, but a larger sample (trial) is needed to evaluate the effect of the intervention definitively. Based on our findings, with an observed power of 0.75 for the within factor (time; η^2^ = 0.79) and 0.24 for the interaction effect (time x group; η^2^ = 0.21), a sample size of at least 50 participants for the intervention group (and potentially 50 participants for the control group) would be needed to conduct a larger/full-scale, adequately powered trial (1 − β = 0.95 with α = 0.05) in this rehabilitation setting. Another limitation is the short follow-up time of three weeks. Future interventions will include a post-rehabilitation follow-up to increase the power of the findings. The use of the HAQ in estimating physical function was chosen to assess physical pain, function, and health in general, but the patients’ social fitness and well-being may have been underrecognized. Nevertheless, the HAQ is one of the most widely used comprehensive, validated, patient-oriented outcome assessment instruments for the evaluation of functional limitations in activities of daily living. A major critical drawback of this study was the rather high protein intake observed in the control group, as they could freely choose what and how much to consume; ideally, strict guidelines regarding what and when to eat should have been given. Finally, relevant covariates, such as the participants’ food habits and lifestyle behaviors before rehabilitation, as well as the exercise type or intensity, were not assessed. Although information extracted from discussions with physiotherapy staff was not quantified, this seems to indicate that most older patients performed low-intensity exercise. While low-intensity physical activity has significant health benefits, moderate- to high-intensity resistance training is recommended to increase muscle mass, strength, and function in older adults [40]. Because of practical constraints, it was not possible to blind our participants or the study assessors for the intervention allocation, which could have influenced our results.

## 5. Conclusions

The current pilot study provides insights into the implementation and outcomes of a randomized, controlled trial using individualized nutrition support in this unique rehabilitation setting. Our findings show that short-term intervention with protein-enriched foods and drinks enabled older patients at risk of malnutrition to increase their protein intake to levels that are higher than their required intake. In these older adults with appropriate protein intakes, dietary protein supplementation did not result in a greater improvement in physical recovery outcomes during short-term inpatient rehabilitation. This might be explained by the low difference in total protein intake between the groups of 0.3 g/kg body weight per day and the small sample size. Although this intervention shows promise in older patients at risk of malnutrition, further research (e.g., a full-scale, randomized, controlled trial with sufficient power) is required to determine the effects on physical function outcomes. Moreover, the observed high prevalence of sarcopenia among geriatric rehabilitation patients highlights the need for enhanced rehabilitation programs that target early screening of malnutrition and sarcopenia in a multidisciplinary approach to ensure that older adults are assessed and treated accordingly.

## Figures and Tables

**Figure 1 healthcare-11-02317-f001:**
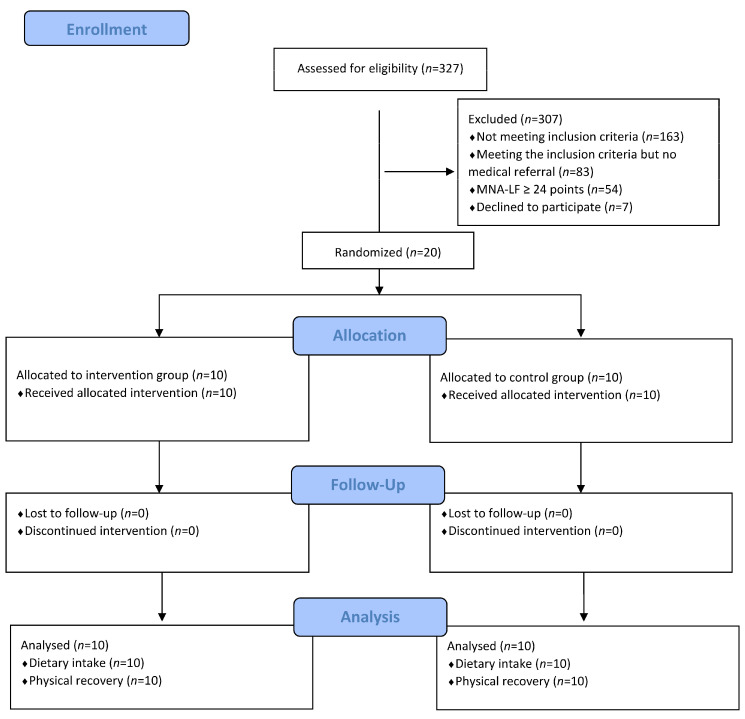
Flow diagram of participant enrollment.

**Figure 2 healthcare-11-02317-f002:**
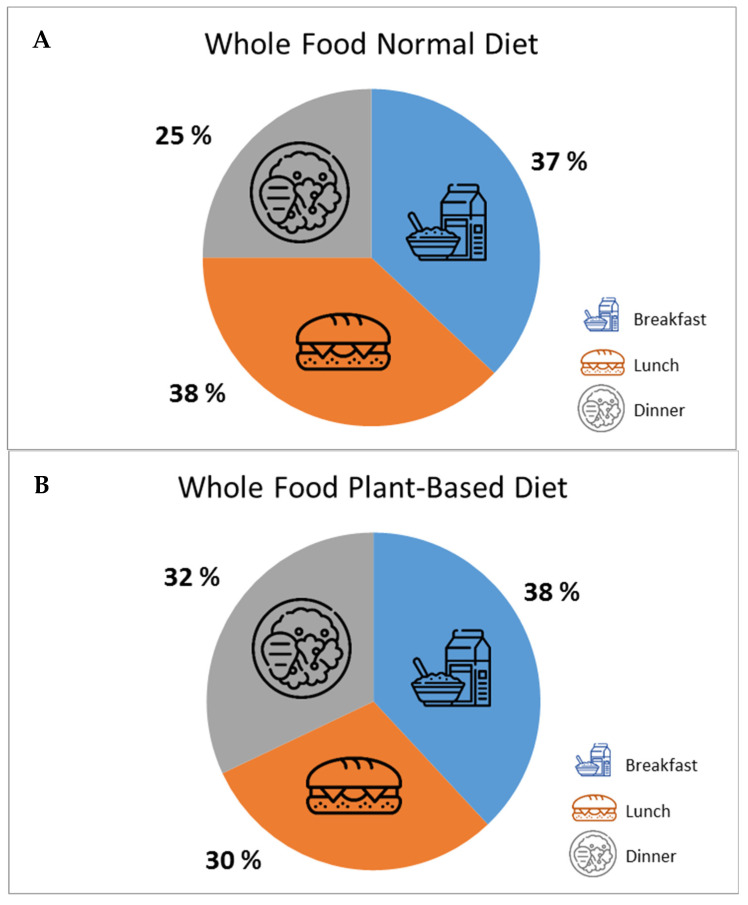
Average protein distribution across self-selected main meals (expressed as a percentage of the total amount of protein provided) in 20 older patients during rehabilitation. (**A**) whole food normal diet, (**B**) whole food plant-based diet.

**Table 1 healthcare-11-02317-t001:** Patients’ characteristics.

	Intervention (*n* = 10)	Control (*n* = 10)	*p*
Age (years)	75.0 ± 6.5	74.2 ± 6.8	0.792
Sex (male/female)	1/9	2/8	0.531
Height (cm)	163.0 ± 8.3	162.6 ± 6.8	0.907
Body weight (kg)	60.4 ± 6.9	62.2 ± 15.0	0.735
BMI (kg/m^2^)	22.9 ± 4.0	23.5 ± 5.9	0.792
RMR (kcal/d)	1198 ± 99	1232 ± 174	0.604
MNA (score)	21.4 ± 2.4	20.9 ± 1.3	0.575
Mobility aids (yes/no)	4/6	4/6	1.000
Medical diagnosis for admission (*n*)			0.819
Arthrosis	1	1	
Diseases of the musculoskeletal system	2	1	
Lower limb injuries	7	8	

Values are means ± standard deviation (SD); numbers and percentages; MNA, Mini Nutritional Assessment. Resting metabolic rate (RMR) was calculated based on gender, body weight, height, and age using the adjusted Harris and Benedict equation [14]: Male: (88.4 + 13.4 × weight in kilograms) + (4.8 × height in centimeters) − (5.68 × age). Female: (447.6 + 9.25 × weight in kilograms) + (3.10 × height in centimeters) − (4.33 × age).

**Table 2 healthcare-11-02317-t002:** Physical-recovery outcomes at admission (t1) and discharge (t2) in the intervention and control group.

	Intervention (*n* = 10) Mean ± SD	*p Time* (t2 − t1) Intervention	Control (*n* = 10) Mean ± SD	*p Time* (t2 − t1) Control	*p Time* Both Groups	*p Group* t1 and t2	*p Interaction* Time × Group
Body weight t1 (kg)	60.4 ± 6.9	0.039 *	62.2 ± 15.0	0.375	0.038 *	0.762	0.487
Body weight t2 (kg)	61.3 ± 6.3		62.6 ± 14.9				
Hand-grip strength t1 (kg)	20.7 ± 9.9	0.746	18.3 ± 5.5	0.444	0.398	0.516	0.652
Hand-grip strength t2 (kg)	20.9 ± 9.1		18.9 ± 4.3				
Prealbumin t1 (mg/dL)	21.1 ± 8.1	0.070 ^#^	23.8 ± 6.1	0.634	0.230	0.739	0.092 ^†^
Prealbumin t2 (mg/dL)	23.6 ± 8.9		23.9 ± 7.1				
Fat-free mass t1 (kg)	43.7 ± 7.1	0.327	42.6 ± 7.9	0.647	0.537	0.862	0.257
Fat-free mass t2 (kg)	43.0 ± 6.8		42.8 ± 8.6				
Body cell mass t1 (kg)	16.2 ± 6.0	0.881	16.4 ± 3.4	0.698	0.982	0.991	0.765
Body cell mass t2 (kg)	16.4 ± 3.7		16.2 ± 4.1				
Fat mass t1 (kg)	17.2 ± 7.8	0.050 *	19.2 ± 10.0	0.923	0.182	0.705	0.110
Fat mass t2 (kg)	18.1 ± 7.3		19.2 ± 9.9				
Phase angle t1 (°)	3.8 ± 1.0	0.958	3.9 ± 0.6	0.521	0.757	0.769	0.853
Phase angle t2 (°)	3.8 ± 0.5		3.8 ± 0.6				
HAQ score t1	0.42 ± 0.32	0.135	0.59 ± 0.28	0.589	0.241	0.256	0.879
HAQ score t2	0.28 ± 0.27		0.48 ± 0.70				

Values are means ± standard deviation (SD); numbers and percentages. HAQ, Health Assessment Questionnaire. * Significant time effect (*p* < 0.05) compared with admission within a group; ^#^ Trend toward significant time effect (*p* < 0.10) compared with admission within a group. ^†^ Trend toward significant intervention effect or interaction of intervention and time effect (*p* < 0.10).

## Data Availability

The data presented in this study are available upon request from the corresponding author.

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
