# Peer review of "Short-Term Effects of Dietary Protein Supplementation on Physical Recovery in Older Patients at Risk of Malnutrition during Inpatient Rehabilitation: A Pilot, Randomized, Controlled Trial"

_healthcare, 2023, doi:10.3390/healthcare11162317_

Round 1
Reviewer 1 Report (Previous Reviewer 1)
I thank the authors for addressing my comments, however, protocol registration is a fundamental aspect of trial research. It is something that should be well-known among academicians and researchers.
The sample size is very small and no information on how they calculated the minimum sample size to detect differences has been provided.
I disagree with the authors in respect of the nature of this trial. Actually, in their rebuttal, the authors stated that the aim of the current manuscript was to assess feasibility, however, the title and aim of the manuscript, as well as the results spoke about efficacy.
Many covariates are missing
Round 2
Reviewer 1 Report (Previous Reviewer 1)
I am satisfied with the authors' reply
This manuscript is a resubmission of an earlier submission. The following is a list of the peer review reports and author responses from that submission.
Round 1
Reviewer 1 Report
The topic is interesting, however, the quality of the study is quite poor.
The total number of participants is 20, very low for addressing any type of research question.
Many fundamental covariates were not assessed. for instance, participants' food habits before hospital admission, the type and intensity of exercise performed during the hospitalization, and as a baseline.
Some specific comments below:
please specify the software used to generate the random number of allocations.
flow chart of inclusion/exclusion participants should be added.
Please, also add a per-protocol analysis (as supplementary material)
did the authors register in advance the protocol of this trial? please add information.
the figures should be moved to the results section.
Author Response
The authors would like to thank Reviewer 1 for evaluating our manuscript and providing expert opinions, which have helped us, improve our work.
The topic is interesting; however, the quality of the study is quite poor. The total number of participants is 20, very low for addressing any type of research question. Many fundamental covariates were not assessed, for instance, participants' food habits before hospital admission, the type and intensity of exercise performed during the hospitalization, and as a baseline.
A: We agree with you. Nevertheless, studies assessing feasibility and efficiency of a combined nutrition and physical therapy in geriatric rehabilitation inpatients remain limited. The current trial provides insights into the implementation and outcomes of a pilot study using individualized nutrition support in this unique rehabilitation setting. We addressed the most salient limitations of our pilot study in the discussion section.
Some specific comments below:
Please specify the software used to generate the random number of allocations.
A: We used the Randomizer for Clinical Trials tool developed at the Medical University of Graz (http://www.randomizer.at/).
Flow chart of inclusion/exclusion participants should be added.
A: Flow diagram of participant enrolment is now presented in Figure 1.
Please also add a per-protocol analysis (as supplementary material).
A: We are sorry for the confusion. A per-protocol analysis was performed as all patients strictly adhered to the protocol, thus, providing an estimate of the true efficacy of an intervention.
Did the authors register in advance the protocol of this trial?
A: Our pilot study presents the research findings of a graduate-level master’s study. The Medical Ethics Research Committee of Innsbruck Medical University gave their approval for this study and keeps an internal clinical trial register. Therefore, we did not register the study in advance as we were not aware of the importance of registering our protocol in advance. This mistake will not happen again in the future.
The figures should be moved to the results section.
A: The figures are now in the results section.
Reviewer 2 Report
Thank you for this review invitation, very interesting and insightful manuscript.
Few comments and suggestions: on the physical functional status, what is the golden standard of functional status check in aging population? Does HAQ is accurate? Suggested to the authors to explain this - important as tool/instrument credibility check as well
On the result, it is a common knowledge to present the outcome of intervention vs control in a table
Author Response
The authors would like to thank Reviewer 2 for evaluating our manuscript and providing expert opinions, which have helped us, improve our work.
Thank you for this review invitation, very interesting and insightful manuscript.
Few comments and suggestions:
What is the golden standard of functional status check in aging population? Is HAQ is accurate? Suggested to the authors to explain this - important as tool/instrument credibility check as well.
A: Thank you for this important comment. In our opinion, there is no golden standard of function status check in aging population as functional status is directly influenced by health conditions, particularly in the context of an elder's environment and social support network. There are several tools, which have been developed and validated, such as the Barthel-Index, the Katz-Index for Activities of Daily Living (ADL) or the Lawton and Brody scale for Instrumental Activities of Daily Living (IADL). In the present study we used the HAQ, which has become a mandated outcome measure for clinical trials in rheumatoid arthritis and in studies of normal aging. The HAQ is one of the most widely used comprehensive, validated, patient-oriented outcome assessment instruments for the evaluation of functional limitations in activities of daily living. Minimal clinically important differences are usually set at 0.22, clinically significant differences are presumed to be at least 0.5. As with any instrument, the HAQ has limitations. The HAQ based on generic, patient-centered dimensions, rather than process measures and is used to assess physical pain, function and health in general, but mental and social well-being are not routinely assessed (e.g., Fries et al. Arthritis Rheum 1980; 23:137). Thus, patients’ social fitness and well-being may be under recognized.
It is a common knowledge to present the outcome of intervention vs. control in a table.
A: Physical-recovery outcomes at admission and discharge in the intervention and control group are now presented in Table 2.
Reviewer 3 Report
Regarding the abstract:
Sufficient? to combact? why is the method mentioned and not what it evaluates in the case of the hand grip? in what sense "all patients"? why is the number of subjects in the aim? reporting the differences in protein intake as a result in absolute terms has not sense but only per kg of body weight, first reporting the average body weight and any differences in weight between the two groups.
About the introduction:
the first part is clear and the bibliographic references are pertinent but the objective is written in a confused way. Moreover, the hypothesi was "that older patients consume protein at intake levels well below the recommended levels during an inpatient rehabilitation stay". But in the methods it is reported that this rehabilitation center already provides a energy and protein-rich hospital menu, what is the point of providing an additional supplement?.
Methods:
what is the starting sample? was the diagnosis of malnutrition made only using MNA? in the nutritional intervention section it is not clear whether completely different meals were constructed from the control group or were enriched foods and drinks provided in addition? why does nutritional intake become an outcome if it is the intervention? do you mean compliance with meals? it would have been useful to assess whether or not each meal was finished and not to compile a three-day dietary protocol, which is not described either.
Results:
a single table with demographic characteristics does not allow a quick understanding of the results. I also find them written in a messy way and no relevance is given to body composition.
Discussion:
there is no real disquisition of the results, moreover the two figures with the relative description are part of the results and not of the discussion. no real conclusion is reached.
I also find it inappropriate to mention oneself in the final part of the conclusions.
Author Response
The authors would like to thank Reviewer 3 for evaluating our manuscript and providing expert opinions, which have helped us, improve our work.
Abstract: Sufficient to combat? Why is the method mentioned and not what it evaluates in the case of the hand grip? In what sense "all patients"? Why is the number of subjects in the aim? Reporting the differences in protein intake as a result in absolute terms has no sense but only per kg of body weight, first reporting the average body weight and any differences in weight between the two groups.
A: Thank you for your suggestion. The abstract has been partially rewritten. However, we decided not to remove the differences in protein intake as a result in absolute terms as other works (e.g., Beelen et al. Clin Nutr 2018) also reported both absolute and relative values of protein intake. Furthermore, studies on protein consumption per main meal reported results in absolute terms (e.g., Weijzen et al. J Nutr Health Aging 2019).
Introduction: The first part is clear and the bibliographic references are pertinent but the objective is written in a confused way. Moreover, the hypothesis was "that older patients consume protein at intake levels well below the recommended levels during an inpatient rehabilitation stay". But in the methods it is reported that this rehabilitation center already provides an energy and protein-rich hospital menu, what is the point of providing an additional supplement?
A: We are sorry for the confusion. Because we did not want to act against the standard advice to consume a protein-rich diet, the control group received a regular protein-rich whole food diet. The rehabilitation center offers protein- and energy-rich menus to older patients during rehabilitation, without the use of protein supplements and drinks. However, even the recommended high protein intake of 1.2 to 1.5 g/kg per day may be too low for functionally limited older patients to improve recovery of muscular strength and physical function, especially in those with chronic diseases (e.g., Bhasin et al. JAMA Intern Med 2018). To overcome older adults’ difficulties in consuming enough protein, we provided an additional supplement.
Methods: What is the starting sample? Was the diagnosis of malnutrition made only using MNA? In the nutritional intervention section it is not clear whether completely different meals were constructed from the control group or were enriched foods and drinks provided in addition? Why does nutritional intake become an outcome if it is the intervention? Do you mean compliance with meals? It would have been useful to assess whether or not each meal was finished and not to compile a three-day dietary protocol, which is not described either.
A: Flow diagram of participant enrolment is now presented in Figure 1. Diagnosis of malnutrition was made using MNA only. The MNA (long-form) is a widely used comprehensive, validated nutrition screening and assessment tool that can identify geriatric patients age 65 and above who are malnourished or at risk of malnutrition. Patients in both groups were free to choose different menus, but only the intervention group received the protein-enriched foods and drinks provided in addition. Protein and energy intake are potential confounders. As such, nutritional intake was measured in conjunction with the study intervention. Poor health, including age-related anorexia, was the major reason. Protein supplements may result in a reduced compliance with meals as protein supplements and drinks are highly satiating. We agree with you that it would have been useful to measure energy and protein intake by the use of visual food protocols or weighing the meals. However, food protocols are hardly integrated into daily clinical practice, which is primarily because the estimation process is time-consuming and is subjective to the perception of the nursing staff that is responsible for the procedure. Therefore, protein and energy intake was estimated based on an average of three 3-day dietary protocols. A checklist of specific food and beverages was used to verify the reported intake, and a visual guide to portion sizes was used to estimate portion sizes.
Results: A single table with demographic characteristics does not allow a quick understanding of the results. I also find them written in a messy way and no relevance is given to body composition.
A: Physical-recovery outcomes (including body composition parameter) at admission and discharge in the intervention and control group are now presented in Table 2.
Discussion: There is no real disquisition of the results; moreover the two figures with the relative description are part of the results and not of the discussion. No real conclusion is reached. I also find it inappropriate to mention oneself in the final part of the conclusions.
A: The figures are now in the results section. In conclusion, dietary protein supplementation increasing the total protein intake by 0.3 g/kg per day during short-term inpatient rehabilitation does not seem to improve physical recovery outcomes. We removed the self-citation in the final part of the conclusions.